# Towards Better Sinomenine-Type Drugs to Treat Rheumatoid Arthritis: Molecular Mechanisms and Structural Modification

**DOI:** 10.3390/molecules27248645

**Published:** 2022-12-07

**Authors:** Cuili Zhang, Shujie Zhang, Jingjing Liao, Zipeng Gong, Xin Chai, Haining Lyu

**Affiliations:** 1School of Medicine, Huanghe Science and Technology College, Zhengzhou 450006, China; 2Artemisinin Research Center, China Academy of Chinese Medical Sciences, Beijing 100700, China; 3State Key Laboratory of Functions and Applications of Medicinal Plants, Guizhou Provincial Key Laboratory of Pharmaceutics, Guizhou Medical University, Guiyang 550004, China

**Keywords:** sinomenine, rheumatoid arthritis, traditional Chinese medicine, autoimmune diseases, inflammatory diseases

## Abstract

Sinomenine is the main component of the vine *Sinomenium acutum*. It was first isolated in the early 1920s and has since attracted special interest as a potential anti-rheumatoid arthritis (RA) agent, owing to its successful application in traditional Chinese medicine for the treatment of neuralgia and rheumatoid diseases. In the past few decades, significant advances have broadened our understanding of the molecular mechanisms through which sinomenine treats RA, as well as the structural modifications necessary for improved pharmacological activity. In this review, we summarize up-to-date reports on the pharmacological properties of sinomenine in RA treatment, document their underlying mechanisms, and provide an overview of promising sinomenine derivatives as potential RA drug therapies.

## 1. Introduction

Rheumatoid arthritis (RA) is a chronic autoimmune and inflammatory disorder characterized by symmetrical pain and swelling of the hands, wrists, feet, and knees. It is associated with severely impaired movement, early death, and significant socioeconomic burden [1]. It arises more frequently in females than in males and is predominantly observed between the ages of 50 and 60 years, with a worldwide prevalence of approximately 5 per 1000 [2]. Non-steroidal anti-inflammatory drugs and disease-modifying anti-rheumatic drugs (DMARDs) are frequently used to treat RA by alleviating inflammation and pain, reducing joint damage, and preserving the structure and function of the joints. The most common conventional DMARDs include methotrexate (MTX), sulfasalazine, and hydroxychloroquine; however, their use is accompanied by adverse events related to their unique action mechanisms [2,3]. For example, MTX, the most frequently used DMARD in clinical practice, can cause stomach upset, mouth sores, and bone lesions. In contrast, sulfasalazine causes changes in blood count, photosensitivity, nausea or vomiting, skin rash, and headaches. Thus, the development of alternative therapies is crucial for the management and treatment of RA.

Complementary and alternative medicines (CAMs) have been widely used to treat RA for thousands of years and confer certain advantages relative to conventional therapies, such as mild toxicity and a lower risk of addiction [4]. CAMs are also a rich source of potential anti-RA agents. *Sinomenium acutum*, a Chinese herbal medicine, has been broadly used as a folk remedy to treat neuralgia and rheumatoid diseases (Figure 1) [5]. Sinomenine (Figure 1C), the main chemical component of this herbal medicine, was first isolated by Ishiwari in the 1920s [6,7] and has since attracted special interest as a potential anti-RA agent, owing to its successful application in traditional Chinese and Japanese medicine to treat inflammatory and rheumatic diseases. Numerous pharmacological and clinical studies have demonstrated that sinomenine possesses extensive therapeutic properties, such as anti-inflammatory, immunosuppressive, and analgesic effects [8]. Sinomenine hydrochloride has been developed in various formulations as an effective and relatively safe drug to treat RA in China [7]. The therapeutic efficacy and mild side-effect profile of sinomenine and combination therapies in RA patients have been demonstrated in multiple clinical trials, and the underlying mechanisms of action have been thoroughly investigated [9,10]. Furthermore, several sinomenine derivatives were synthesized to improve their anti-inflammatory, immunosuppressive, and analgesic effects [11]. Herein, we present an up-to-date review of the current literature on the multitarget molecular mechanisms of sinomenine, as well as its structural modifications that improve RA treatment and management.

## 2. Direct Targets of Sinomenine for Treating RA

To explore the drug targets of sinomenine, Guo et al. used network pharmacology to search for potential targets and signaling pathways through which sinomenine treats RA [12]. Consequently, 16 potential targets, including HSP90AA1, MAP3K3, F13A1, CTNNB1, HDAC6, ERBB2, and KIT, were identified in the sinomenine and RA intersection target networks and were subsequently validated via enrichment analysis. Although further experimental validation is required, their reliability in targeting RA has been reported. For example, HSP90AA1 is involved in arachidonic acid metabolism pathways, a major mediator of many inflammatory diseases, and is thus considered an “RA-specific” gene [13]. MAP3K3, another well-studied RA identification gene, is also involved in the immune response [14]. The human coagulation factor XIII A subunit gene F13A1 plays an essential role in early innate immune responses by modulating inflammation [15], whereas CTNNB1 plays a critical role in inhibiting the proliferation and inflammatory response of fibroblast-like synoviocytes (FLSs) in RA pathogenesis [16]. Moreover, according to enrichment analysis results, these targets could ameliorate RA by inhibiting synovial hyperplasia, angiogenesis, and cartilage destruction [12]. Recently, Chen et al. used an activity-based protein profiling approach to elucidate the direct anti-inflammatory effect of sinomenine on the murine macrophage RAW 264.7 [17]. First, a clickable probe was synthesized and confirmed to retain an anti-inflammatory function similar to that of sinomenine. Subsequently, in-gel fluorescence analysis revealed that probe-induced labeling was highly influenced by pretreatment of the cell lysate with sinomenine, indicating that the probe-labeled proteins were sinomenine targets. Using this probe, 2755 proteins were identified, of which 92 were considered potential sinomenine targets. Bioinformatic analysis revealed that the anti-inflammatory effects of sinomenine follow a poly-pharmacological mode of action. The top 20 potential protein targets were selected to illustrate their effects on inflammation. Knockdown of certain genes, such as Ripk 3, Ptges3, Prdx4, and Dbnl, decreased inflammatory cytokine levels, including interleukin (IL)-1β and IL-6, as well as tumor necrosis factor (TNF)-α in lipopolysaccharide (LPS)-stimulated cells. ADH5, Ddx27, Dld, Myh11, Snx5, Syne3, and Utp18 knockdown resulted in significant inflammation following the induction with LPS, suggesting their roles in attenuation of inflammation [17].

The above results indicate that sinomenine could be a multitarget anti-RA drug. Additionally, Chen et al. revealed that a selective reduction in the carbon–carbon double bond of α,β-unsaturated ketones significantly reduced the ability of sinomenine to inhibit inflammatory cytokine expression, indicating that the α,β-unsaturated ketone group is the key active group and may covalently bind to the target proteins [17]. Further investigation should be conducted to elucidate the sites and mechanisms of binding of sinomenine to these target proteins and how they correlate with RA regulation.

## 3. Immunosuppressive and Anti-Inflammatory Effects of Sinomenine

Inflammation is an immune-related protective response that follows several events, such as infections, exposure to toxins, and post-ischemic changes [18]. As an anti-RA drug, the immunosuppressive and anti-inflammatory properties of sinomenine have been previously documented [10]; sinomenine suppresses lymphocyte activity by inducing apoptosis, inhibiting proliferation, and regulating the Th1/Th2 imbalance [19,20,21] (Figure 2). In particular, sinomenine-induced apoptosis in CD4+ primary lymphocytes is mediated through the caspase 3-dependent pathway rather than the B-cell lymphoma-2 protein family [21]. Furthermore, the potential suppressive effects of sinomenine on dendritic cells (DCs) and monocytes/macrophages have been documented; sinomenine was reported to moderate DC differentiation, maturation, and functionality and improve antigen uptake by LPS-stimulated DC [22] (Figure 2). Zhao et al. reported that sinomenine could decrease the antigen-presenting activity of DCs by suppressing the NF-κB pathway by decreasing I-κBα phosphorylation rather than attenuating RelB and p38SAPK expression [23]. He et al. reported that sinomenine induced RAW264.7 apoptosis by activating the extracellular signal-regulated protein kinase (ERK), which was accompanied by increased p27 and Bax expression in the apoptotic macrophages [24]. Furthermore, Ou et al. reported that sinomenine markedly inhibited activated human monocytic THP-1 cell migration and invasion in a dose-dependent pattern [25]. A possible mechanism for this phenomenon is through reduced expression of matrix metalloproteinases 2 and 9, which highly correlates with attenuated CD147 activity.

Novel insights into the molecular mechanisms underlying the immunomodulatory and anti-inflammatory functions of sinomenine have recently been reported. In 2015, Tong et al. reported that sinomenine modulated Th17 and regulatory T-cell frequency in intestinal lymph nodes and yielded tissue-selective lymphocyte trafficking from the gut to the joint, thereby suppressing collagen-induced arthritis (CIA) [26] (Figure 2). In 2016, they also reported that sinomenine was an aryl hydrocarbon receptor (AhR) agonist that can induce AhR target gene expression, promote the dissociation of the AhR/Hsp90 complex and the nuclear translocation of AhR, induce XRE reporter activity, and facilitate AhR/XRE binding. In a CIA mouse model, the antiarthritic effect of sinomenine was largely diminished by the AhR antagonist resveratrol [27].

Microsomal prostaglandin E 2 synthase 1 (mPGES-1) catalyzes the terminal step of prostaglandin E2 (PGE2) biosynthesis and has been associated with various types of human diseases, including RA. Thus, mPGES-1 is considered a potential anti-RA drug target [28]. Zhou et al. reported that sinomenine reduced PGE2 levels via selective suppression of mPGES-1 expression without affecting the formation of prostacyclin and thromboxane [29]. In the edema rat model and CIA DBA mouse model, mPGES-1 protein expression was downregulated following sinomenine treatment in inflamed paw tissues, suggesting that sinomenine can selectively inhibit mPGES-1. In a follow-up study, the exact mechanism was investigated [30]; the DNA demethylating agent 5-AzaC was reported to reverse the inhibition of sinomenine on mPGES-1. Sinomenine selectively increased methylation levels at the specific GCG sites in the mPGES-1 gene promoter, and 5-azacytidine pretreatment suppressed this effect.

In 2018, Liu et al. measured several clinical indices, inflammatory cytokine secretion, and the disease activity score (DAS28) in RA patients treated with sinomenine to explore its anti-inflammatory mechanism in a clinical setting [31]. The results demonstrated that sinomenine can regulate eotaxin-2, GM-CSF, IL-1α, IL-1β, IL-6, IL-10, IL-12 p40, KC (CXCL1), MCP-1, M-CSF, RANTES, and TNF-α secretion and reduce RA activity and the DAS28 score. Moreover, the CD14+CD16+ blood monocyte count was decreased. The anti-RA effect of sinomenine was reported to be mediated by gut-sourced vasoactive intestinal polypeptide (VIP), which is generated through α7nAChR activation [32]. Orally administered sinomenine improved systemic inflammatory conditions in CIA rats in a manner similar to the administration of nicotinic receptor antagonists, including α7nAChR antagonists but not muscarinic receptor antagonists. In addition, elevated VIP levels in the small intestine and serum of rats negatively correlated with joint destruction. The anti-RA effect of oral sinomenine can be summarized in the following order: (1) stimulation of α7nAChR; (2) activation the PI3K/Akt/mTOR pathway; and (3) generation of the anti-inflammatory neuropeptide VIP in the small intestine, which enters systemic circulation to regulate the inflammatory response pathway. The role of α7nAChR in the inflammation reflex and the potential mechanism of sinomenine in the modulation of α7nAChR have recently been elucidated [33]. In RAW264.7 cells treated with α7nAChR shRNA and stimulated with LPS, multiple classical M1 markers, such as IL-6, iNOS, and TNF-α, were downregulated, whereas M2 markers, such as Arg-1, Fizz1, and IL-10, were upregulated. Sinomenine suppressed Egr-1 and p-ERK1/2 expression in the LPS-induced RAW264.7 cells, and α7nAChR expression was inhibited by U0126, which in turn reduced the expression of Egr-1 and p-ERK1/2. This supports the notion that sinomenine downregulates α7nAChR through the α7nAChR/ERK/Egr-1 feedback pathway, which subsequently inhibits the polarization of the M1-type macrophage and resolves inflammation.

Macrophage migration is known as a fundamental process in immune responses, whereby monocytes leave the blood and differentiate into macrophages within the inflammation site of tissues [34]. Gao et al. recently reported that sinomenine can reduce the population of RAW264.7 cells migrating toward inflammatory paws and block the migration of bone-marrow-derived macrophages. The absence of macrophage migration after the depletion of circulating and peripheral macrophages reduced the inflammatory response severity. Research has revealed that sinomenine can inhibit macrophage migration by activating the Src/FAK/P130Cas axis [35].

## 4. Antiarthritic Effect of Sinomenine

RA is clinically characterized by physical disability, which is caused by persistent joint inflammation and destruction [36]. Bone tissue destruction is primarily attributed to abnormal activation and/or enhanced differentiation of osteoclasts, which are ultimately differentiated into monocytes and macrophages [37]. Thus, osteoclast regulation is considered a potential therapeutic target in RA, and in recent years, the effects of sinomenine on these cells have been extensively studied. In 2014, He et al. revealed that sinomenine inhibited mature osteoclast viability in a dose- and time-dependent pattern [38]; however, it had no significant effect on undifferentiated RAW264.7 cell viability. Actin ring formation, a prerequisite for osteoclast bone resorption, was also impaired following sinomenine treatment. Further mechanistic studies revealed that sinomenine can inhibit osteoclast survival by inducing mature osteoclast apoptosis, at least in part by activating caspase 3. In another study, the inhibitory effect of sinomenine on osteoclastic-specific marker gene expression was observed during LPS-stimulated osteoclast differentiation and survival [39]. Sinomenine inhibited AP-1, NFAT, and NF-κB activation and decreased intracellular Ca(2+) levels and MAPK p38 phosphorylation. In upstream signaling, the expression of TLR4 and TRAF6 was reduced during osteoclast differentiation; however, TLR4, rather than TRAF6, was expressed in the osteoclast survival pathway. Further studies revealed that sinomenine inhibited the PGE2-induced increase in the OPG/RANKL ratio, suppressing osteoclastogenesis in RAW264.7 and serving as a promising proinflammatory mediator that regulates the immunosuppressive function of mesenchymal stem cells [40].

FLSs are activated in RA joints and play an essential role in synovial intimal lining destruction by producing cytokines directly involved in chronic inflammation and joint destruction [41]. In 2005, the effects of various sinomenine doses on pleiotropic cytokine expression in adjuvant arthritis rat models were evaluated in vitro. At concentrations ranging from 30 to 120 μg/mL, sinomenine reduced IL-1β and TNF-α expression in synoviocytes in a concentration-dependent pattern (*p* < 0.05). The inhibitory effect on NF-κB activity was mediated by upregulation of IκB-α expression [42]. Liao et al. recently reported that sinomenine significantly inhibited IL-6 and IL-33 secretion and ROS generation in synovial fibroblasts derived from RA patients. Further investigation indicated that sinomenine can inhibit p62 phosphorylation at Thr269/Ser272 and Ser349 sites to activate the Keap1-Nrf2 pathway, resulting in protection from bone destruction and exerting anti-RA effects [43].

In the previous section, we discussed how intestinal α7nAChR activation could be pivotal in ameliorating arthritis following sinomenine treatment [32]. The important receptor protein α7nAChR, which attenuates unrestrained inflammation within the cholinergic anti-inflammatory pathway, was also detected in the synovium of RA patients. Yi et al. reported that sinomenine inhibited the proliferation of FLSs isolated from AIA rat synovial tissues and reduced ERK1/2 phosphorylation, in addition to decreasing TNF-α induced α7nAChR and Egr-1 expression (*p* < 0.05), delineating a new correlation between α7nAChR activation and FLS proliferation [44]. In a follow-up study, three years later, the same group reported that α7nAChR also mediated adenosine A2A receptor (A2AR) expression. Studies focused on the mechanisms of action showed that sinomenine can upregulate A2AR and cAMP and inhibit NF-κB activation to attenuate arthritis through α7nAChR [45].

In addition to FLS activation, the loss of phenotypic stability of articular chondrocytes is also involved in RA-induced joint destruction [46]. Sinomenine was reported to treat LPS-injured chondrogenic ATDC5 cells and was shown to significantly attenuate ATDC5 cell damage by reducing miR-192 expression and subsequently suppressing MAPK and NF-κB and activation [47]. Restoring the expression of miR-192 significantly impeded the protective effects of sinomenine, indicating that sinomenine blocked MAPK and NF-κB pathways in a miR-192-dependent pattern. In a similar study, Wu et al. reported that sinomenine inhibited IL-1β-induced inflammation and cartilage destruction by upregulating the Nrf2/HO-1 pathway and suppressing NF-κB signaling activity in cartilage cells in mice [48]. In vivo experiments revealed that sinomenine also attenuated cartilage destruction in an osteoarthritis mouse model, suggesting that sinomenine confers a protective effect against osteoarthritis progression.

As a key molecular player in synovial tissue, myeloid differentiation primary response 88 (MyD88) cooperates with NF-κB activity to amplify the production of inflammatory mediators in RA [49]. Ramirez-Perez et al. recently reported that targeting of MyD88 an significantly downregulate the systemic inflammatory mediators and modulate pathological processes in RA patients [50]. Mu et al. reported that sinomenine markedly decreased MyD88 expression, leading to the suppression of the inflammation response and progression of joint destruction in AIA rats [51]. These results further highlight the protective effects of sinomenine in preventing joint destruction and deformation in patients with RA.

## 5. Antinociceptive Effect of Sinomenine

In addition to its immunosuppressive and anti-inflammatory effects, sinomenine also shows potent analgesic efficacy in RA management, such as by attenuating pain by reducing physiological pain indicators. Moreover, sinomenine treats chronic pain by suppressing inflammation and modulating neuroplasticity [8]. In 2013, Gao et al. systematically determined the analgesic effects of sinomenine in rodent models [52]. Sinomenine exhibited moderate antinociceptive effects in hot-plate and tail-flick tests, as well as in an inflammatory pain mouse model induced by carrageenan. It was reported to alleviate both cold and mechanical allodynia following spinal cord or peripheral nerve injury. Notably, the analgesic effect was not accompanied by side effects, nor was it reversed by naloxone, an opioid receptor antagonist. In another study, the antinociceptive activity of sinomenine in a rat postoperative pain model was described [53]. Sinomenine exhibited concentration-dependent antinociceptive activity in rats following surgery, which lasted 4 h and did not produce tolerance; however, this effect could be blocked by bicuculline, a gamma-aminobutyric acid type A (GABA_A_) receptor antagonist. The cellular mechanism underlying the peripheral analgesic efficacy of sinomenine in a formalin-induced acute inflammatory pain mouse model was recently reported [54], demonstrating that sinomenine can reduce paw-licking behavior in the first and second phases, as well as c-Fos protein expression in the dorsal horn of the spinal cord. A patch-clamp study of dorsal root ganglia neurons revealed that sinomenine stimulated spike threshold dynamics and threshold current intensity, decreased the frequency of the firing of action potentials, and suppressed voltage-gated Na+ currents in a dose-dependent pattern, indicating that sinomenine confers a peripheral antinociceptive effect by suppressing voltage-gated Na^+^ currents (Figure 2).

## 6. Novel Sinomenine-Derived Anti-RA Agents

Owing to the broad spectrum of pharmacological activities of sinomenine in treating RA, numerous attempts have been made to synthesize sinomenine-based derivatives with enhanced anti-inflammatory, immunosuppressive, and analgesic properties. Tang et al. systematically reviewed the chemical modifications and various bioactivities of reported sinomenine derivatives in 2018 [11], and Ng et al. reviewed the representative synthetic strategies for accessing various categories of sinomenine derivatives in 2020, with a focus on the chemoselectivity and stereoselectivity within the reaction pathways [55]. In this section, we highlight the sinomenine derivatives with superior anti-RA activity or promising RA treatment potential. Their structures are depicted in Figure 3, and their detailed activity data are summarized in Table 1. The potential structure–activity relationship (SAR) of sinomenine and its derivatives for the treatment of RA is also discussed.

### 6.1. Sinomenine Derivatives with Enhanced Anti-RA Effects

Introducing effective substituent groups is the most convenient strategy for the structural modification of small molecules. In 2011, Wu et al. prepared a series of C-1-substituted sinomenine derivatives via electrophilic substitution [56]. Among them, 1-formyl-sinomenine (compound **2**) showed the most potent inhibitory effects on mouse ear swelling induced by croton oil and was more active against IL-2 release in rat spleen cells compared to sinomenine. In 2015, Zhao et al. synthesized a series of 1-hydroxymethyl ester-4-O-ether derivatives of sinomenine to enhance its anti-inflammatory efficacy [57]. Compound **3** exhibited the most potent anti-inflammatory activity, and, similar to compound **3**, another two 1-hydroxymethyl sinomenine derivatives (compounds **4** and **5**) exhibited increased inhibition of ear swelling and paw edema compared to sinomenine, which were long-lasting effects [58]. Using a capillary-based microfluidic system, two novel series of derivatives of sinomenine were synthesized [59]. The inhibitory effects of these compounds on NF-κB signaling activation were determined in vitro. Compound **6**, which was substituted by a long fatty chain, showed enhanced activity compared to sinomenine. Intraperitoneal administration of this compound markedly attenuated edema levels at 6 h, with efficiency lasting 18 h. Recently, N-demethyl-sinomenine (compound **7**), a sinomenine metabolite, was evaluated for chronic pain conditions. The potency of N-demethyl-sinomenine in reducing mechanical allodynia induced by complete Freund’s adjuvant was slightly better than that of sinomenine. Moreover, it maintained this efficiency without generating a carry-over effect during repeated treatment; however, its effect was almost completely blocked by pretreatment with the GABA_A_ receptor antagonist bicuculline [60]. In 2012, Teng et al. synthesized a series of novel sinomenine homodimers to explore more active molecules with better anti-inflammatory effects [61]. Compound **8** inhibited NO, IL-6, and TNF-α production more potently than sinomenine, and in vivo assays demonstrated that it also markedly relieved inflammatory symptoms and mediated bone loss and joint destruction in CIA mice [62]. Furthermore, it considerably decreased IL-1β, IL-6, and TNF-α levels in the serum of CIA mice and markedly reduced NF-κB p65 phosphorylation and nuclear translocation in FLSs [62]. To improve the therapeutic efficiency of sinomenine, Yan et al. prepared hundreds of sinomenine derivatives by introducing nitrogen-containing heterocyclic systems [63]. Among them, compound **9** suppressed Th17 cell differentiation and improved inflammatory symptoms. The rodent model of experimental autoimmune encephalomyelitis (EAE) suggested that compound **9** possessed an improved ability to reduce T-cell responses and ameliorate EAE compared to sinomenine. To improve the immunoregulatory activity of sinomenine, Zhou et al. reported a series of new sinomenine–pyrazine hybrids, most of which showed better inhibitory effects of TNF-α than sinomenine. Among them, compounds **10** and **11** exhibited up to 99% inhibition [64]. Sinomenine-4-hydroxy-palmitate (compound **12**) was recently reported as a sinomenine derivative [65]. By introducing a palmitic acid group on the 4-OH of sinomenine, compound **12** increased the survival rate of LPS-treated mice compared to sinomenine and decreased the expression levels of inflammatory cytokines, including IL-1β, IL-6, and TNF-α. A molecular mechanism study revealed that compound **12** exerted anti-inflammatory effects by inhibiting LPS-induced phosphorylation of p38, AKT, and STAT1, resulting in ERK1/2 activation. Additionally, it may promote the reprogramming of macrophages toward an M2-like phenotype via the Akt and p38 MAPK signaling pathways. Gao et al. recently synthesized a series of amino-acid- and nitrogen-based heterocyclic derivatives to enhance the anti-inflammatory effects of sinomenine [66]. Among them, compound **13** demonstrated enhanced anti-inflammatory activity in both the LPS-induced RAW264.7 inflammatory cell model and the carrageenan-induced paw edema mouse model. Moreover, compound **13** exhibited peripheral analgesic effects by significantly decreasing the number of acetic-acid-induced writhes—an effect that was even more potent than that of the positive Voltaren group. 

In addition to the structural modification of sinomenine, a drug delivery system was designed and developed to improve the targeting of sinomenine to inflamed tissue for the treatment of RA. Zhang et al. recently reported two novel synovial-targeting peptide–sinomenine conjugates as potential RA-targeted therapeutics [67]. The linear and cyclic forms of the synovial homing peptide (CKSTHDRLC) were conjugated to the hydroxyl group of sinomenine via a six-aminocaproic acid linker. In the estimation of their in vitro biostability and hemolysis activity, as well as their in vivo therapeutic efficacy and biodistribution tests, the cyclic peptide-conjugated sinomenine (**14**) exhibited exceptional stability in both mouse serum and joint homogenate and selectively accumulated in the inflamed joint sites, which alleviated acute inflammation and reduced histological signs of inflammation in the AIA mice.

### 6.2. SAR Analysis of Sinomenine Derivatives for RA Treatment

To date, various sinomenine derivatives have been developed to improve anti-RA efficacy, as well as other bioactivities. According to the review by Tang et al. [11], as well as SAR analysis of compounds **2**–**14**, the C-1, C-4-OH, and the N atoms are among the promising sinomenine modification sites. Minor modifications to these sites, such as the introduction of ester groups to C-1 and C-4-OH (**2**–**6**, **12**, **13**) or demethylation of N-methyl (**7**), could significantly enhance the anti-RA effects of the mother molecule. However, in most cases, minor modifications appear to change only the anti-RA potency of sinomenine rather than its therapeutic mechanism. For example, sinomenine has been previously reported to exert GABAergic analgesia [53]; its N-methyl derivative (compound **7**) showed a similar effect, with increased potency and fewer side effects [60]. In addition to minor modifications, the synthesis of dimers (**8**, **12**–**14**) or hybrids (**9**–**11**) of sinomenine has been shown to be an effective strategy for the development of improved anti-RA agents. The potential mode of action of these newly synthesized sinomenine derivatives is largely unknown. For example, compounds **9**–**11** are sinomenine–pyrazine hybrids, and their anti-RA efficacy was remarkably improved according to the observation of the expression level of various cytokines and signaling pathways. However, as previously mentioned, a medicinal chemistry case study suggested that the α, β-unsaturated ketone group of sinomenine is the key reactive site for its pharmaceutical effects [17]. However, compounds **9**–**11** remained active, despite the removal of this group from their structures (Figure 3); as a result of this finding, the therapeutic targets of sinomenine in RA remain in question. 

## 7. Conclusions

Sinomenine, the principal component of *S*. *acutum*, has been used to treat RA, as well as related immune disorders, for several decades. Its potent therapeutic efficacy and few side effects in RA patients have been clinically validated; however, the underlying mechanism remains poorly understood. In this review, we summarized the reported literature on the potential therapeutic targets and molecular mechanisms of sinomenine as an anti-RA drug, revealing its multiple effects in RA, including immunoregulation, anti-inflammatory effects, alleviation of pain, and improved bone metabolism (Figure 2). Additionally, several promising sinomenine-based anti-RA drug leads have recently been documented (Figure 3), identifying additional potential candidates for the development of innovative anti-RA drugs. Furthermore, sinomenine showed mild side effects such as histamine release-like anaphylactoid reactions (HRARs) in some cases in clinical trials [68]. Therefore, further studies should be conducted with a focus on documenting sinomenine-induced HRARs. Beyond the treatment of RA, sinomenine has also exhibited novel pharmacological actions, such as antitumor capabilities [69], as well as protective effects on cerebral or myocardial ischemia–reperfusion injury, hypertension, heart failure, autoimmune myocarditis, and arrhythmia [7]. The development of alternate uses for existing drugs improves therapeutic options in patient care, and based on the diverse pharmacological activities discussed above, the clinical application of sinomenine in cancer and cardiocerebrovascular diseases should be considered.

## Figures and Tables

**Figure 1 molecules-27-08645-f001:**
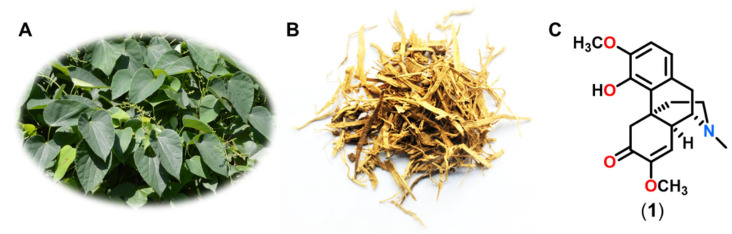
Traditional use of *S*. *acutum* and clinical application of its main active component, sinomenine. (**A**) Plant *S*. *acutum*; (**B**) dry rattan stem of *S*. *acutum*, the picture is reproduced with permission; (**C**) structure of sinomenine.

**Figure 2 molecules-27-08645-f002:**
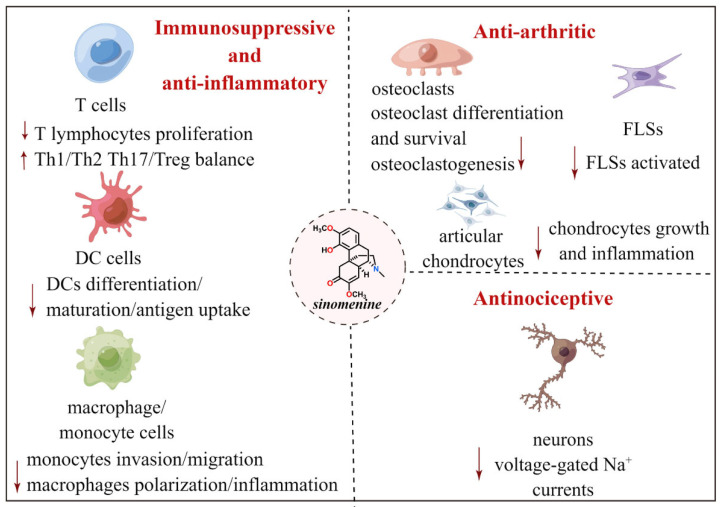
Molecular mechanisms of sinomenine for treatment of RA. Sinomenine exhibits its therapeutic effects on RA by modulating several key immune responses and inflammatory pathways, attenuating inflammation, alleviating pain, and inhibiting bone and joint destruction.

**Figure 3 molecules-27-08645-f003:**
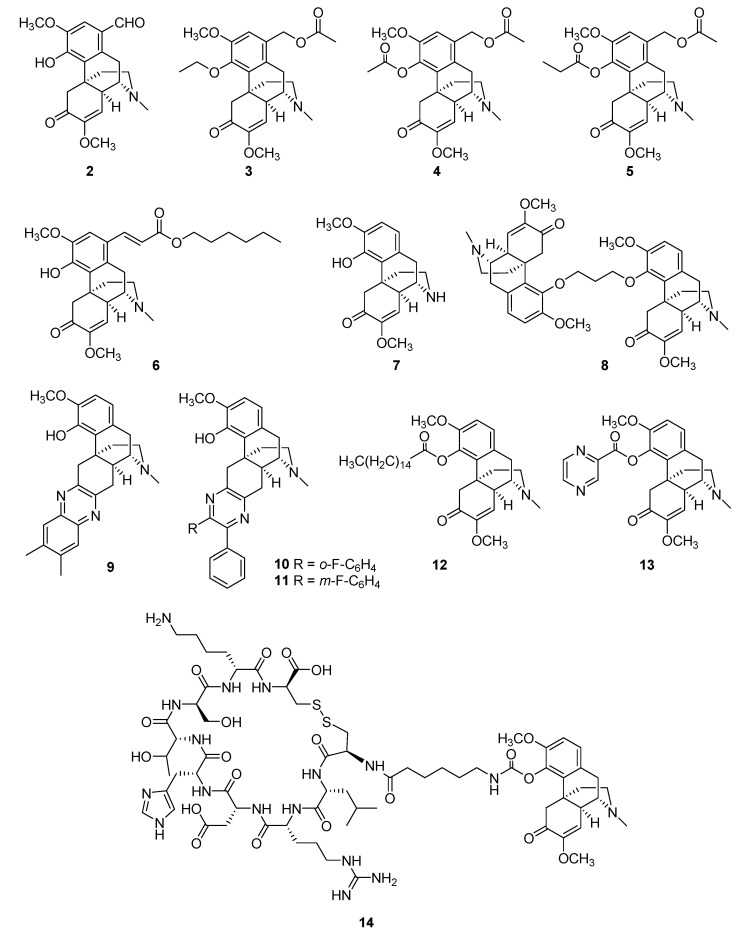
Structures of promising anti-RA sinomenine-based derivatives and conjugates.

**Table 1 molecules-27-08645-t001:** The anti-RA activities of sinomenine derivatives.

Compound	Cell Model and Efficacy	Animal Model	Potential Targets or Pathways	Cytotoxicity or Side Effects	Ref.
**2**	rat splenocytes	croton oil-induced ear edema	IL-2	— ^1^	[56]
**3**	LPS-induced RAW264.7 cells	dimethylbenzene-induced ear edema, carrageenan-induced paw edema	IL-1β, TNF-α	—	[57]
**4**, **5**	LPS-induced RAW264.7 cells	dimethylbenzene-induced ear edema, carrageenan-induced paw edema	IL-1β, TNF-α	no obvious cytotoxicity in peritoneal macrophages at concentrations below 0.02 mg/mL	[58]
**6**	TNF-α-induced mouse embryonic fibroblasts	carrageenan-induced paw edema	NF-κB	cytotoxicity in mouse embryonic fibroblasts, IC_50_ = 38.9 μM	[59]
**7**	—	chronic constriction injury model, complete Freund’s adjuvant-induced mechanical allodynia	GABA_A_ receptor	no sedation or allergic reactions at a dose of 80 mg/kg	[60]
**8**	LPS-induced RAW264.7 cells, NO production inhibition, IC_50_ = 15.2 μM (sinomenine concentration >200 μM); fibroblast-like synovial cells (FLSCs)	LPS-induced septic shock model, collagen-induced arthritis	TNF-α, IL-6, NF-κB, IκBα; not the MAPK pathway	cytotoxicity in RAW264.7 cells, IC_50_ = 41.1 μM (sinomenine, IC_50_ >200 μM); no inhibition of FLSC growth at 30 μM	[61,62]
**9**	Bone-marrow-derived dendritic cells	EAE model	TNF-α, IL-6		[63]
**10**, **11**	LPS-induced peritoneal macrophages, 99% inhibition of TNF-α activity at 10 μM (sinomenine, 11% inhibitory rate)	—	TNF-α	—	[64]
**12**	LPS-induced ANA-1/peritoneal macrophages	LPS-induced endotoxemia model	P38/AKT or STAT1 pathways	—	[65]
**13**	LPS-induced RAW264.7 cells, NO production inhibition, IC_50_ = 30.28 ± 1.70 μM (sinomenine, IC_50_ = 70.86 ± 1.00 μM)	carrageenan-induced paw edema, acetic-acid-induced writhing test	NF-κB, iNOS	no obvious cytotoxicity in RAW264.7 cells	[66]
**14**	—	carrageenan-induced paw edema, AIA model	—	no hemolytic activity	[67]

^1^ Not mentioned in the literature.

## Data Availability

Not applicable.

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
