# Peer review of "Towards Better Sinomenine-Type Drugs to Treat Rheumatoid Arthritis: Molecular Mechanisms and Structural Modification"

_molecules, 2022, doi:10.3390/molecules27248645_

Round 1
Reviewer 1 Report
This review by Zhang et al. summarizes recent developments in the field's understanding of the polypharmacological mechanism of the natural product sinomenine and derivatives for the treatment of rheumatoid arthritis. In terms of scope, it appears to cover developments since a similar 2018 review on the topic by Tang et al.
This review is topical, nicely written and well-organized. The discussion of recent developments in the mechanism (biology) is much stronger than the discussion of recently disclosed sinomenine derivatives (chemistry). The chemistry portion is very superficial in its content, as summarized below:
- There is no attempt made to present the rationale behind any of the structural modifications of the presented simonene derivatives. The article also fails to cite Ng's excellent 2020 review on the topic, which is a pertinent and important reference given the lack of such analysis here: 10.1039/D0QO00785D
- There is no attempt to glean or communicate SAR from the presented simonene derivatives
- There is no attempt to summarize or utilize any prior SAR observations from the Tang et al. review to arrive at a comprehensive understanding of simonene SAR to-date. Such a global analysis would be arguably more useful to the reader than just the isolated structures shown here.
- Several recently-reported 4-hydroxy-substituted analogues are curiously omitted, for example:
- sinomenine-4-hydroxy-palmitate (C16) as described in Ni et al.'s 2021 article: 10.1177/2058738421102678
- sinomenine-4-amino acid derivatives described on Guo et al.'s 2022 article: 10.1039/D2RA05558A
- sinomenine-4-hyroxy peptide conjugates described in Zhang et al.'s 2022 article: 10.1016/j.ijpharm.2022.121628
Other minor points:
- It is not clear what is meant by "multifunctional" in line 275. What are the multiple functions being alluded to by use of this word?
- As a minor note, the structure of the sinomenine core is drawn awkwardly in Figures 1, 2 and 3. First, the perspective is incorrect at the point where the piperidinyl and cylohexyl rings "intersect". Based on the "back" stereochemistry of its other ring substituents, the CH2-CH2 bond of the piperidinyl ring should be moved "behind" the CH2-CH2 bond of the cyclohexyl ring it is intersecting, instead of in front as it is currently depicted. In addition, the dashed stereocenters at the tertiary amine should be reversed, such that they originate from carbon and not nitrogen
In summary, I would consider accepting for publication after improvements to the medicinal chemistry discussion and figures are made, and after the aforementioned recent articles (and any others) are properly covered so as to make this a complete and comprehensive "up-to-date" review as described in the manuscript.
Reviewer 2 Report
In the manuscript, the authors summarize up-to-date reports on the pharmacological properties of sinomenine in treating RA, document its underlying mechanisms, and provide an overview of sinomenine derivatives. Considering the overall results, the article is suitable for publication in Molecules with some revisions, as reported below.
1. The review prioritizes sinomenine, and the title creates the expectation that information on different derivatives compounds from sinomenine will be found. Thus, I suggest changing the title of the review.
2. Only in item 6, "Novel sinomenine-derived anti-RA agents," are senomine derivatives mentioned. However, only 11 compounds were described by the authors because they were the ones that showed the best anti-AR results. It would be interesting if the authors made available, as supplementary material, a table with all the compounds that were evaluated with anti-RA in the articles used to carry out this review and identify the targets that were tested. Cytotoxicity data should also be included, if any.
3. The authors state that the synonymous alkaloid was isolated for the first time in 1920. Which period did the bibliographic survey for this review include?
Reviewer 3 Report
The present review article deals the molecular mechanisms and structural modifications of sinomenine type drugs for the treatment of rheumatoid arthritis. This review explains the pathological conditions involved in the rheumatoid arthritis and how the sinomenine and their derivatives prevent/cure the RA. Really it is an interesting work and the review covers latest information about the selected compound. The objective was explained well and all the possible mechanisms were discussed in an extended manner. The review was concluded well. Moreover the manuscript was written well with good English. I do not find any grammatical or spelling mistakes throughout the manuscript.
Author Response
Thanks very much for your comments.
Round 2
Reviewer 1 Report
The revisions have nicely addressed all prior noted concerns. The revised manuscript is much improved and I recommend it be accepted for publication.